# The Effects of Using Partial or Uncorrected Correlation Matrices When Comparing Network and Latent Variable Models

**DOI:** 10.3390/jintelligence8010007

**Published:** 2020-02-15

**Authors:** Dennis McFarland

**Affiliations:** National Center for Adaptive Neurotechnologies, Albany, NY 12208, USA; mcfarland@neurotechcenter.org

**Keywords:** intelligence, process overlap theory, psychometric network analysis, latent variable modeling

## Abstract

Network models of the WAIS-IV based on regularized partial correlation matrices have been reported to outperform latent variable models based on uncorrected correlation matrices. The present study sought to compare network and latent variable models using both partial and uncorrected correlation matrices with both types of models. The results show that a network model provided better fit to matrices of partial correlations but latent variable models provided better fit to matrices of full correlations. This result is due to the fact that the use of partial correlations removes most of the covariance common to WAIS-IV tests. Modeling should be based on uncorrected correlations since these represent the majority of shared variance between WAIS-IV test scores.

## 1. Introduction

The structure of cognitive test batteries, such as the WAIS-IV or the WJ-IV are typically modeled with factor analytic procedures ([5]) or covariance structural modeling ([2]). These methods account for the covariance between test scores with latent variables. Network models have been proposed as solutions to the problem of modeling psychological constructs such as the symptoms of psychopathology ([4]) and more recently, cognitive test scores ([10]; [15]). The network models proposed by these investigators do not make use of latent variables, but rather model the structure of test batteries in terms of manifest variables (i.e., observed test scores).

Psychometric network models represent correlation matrices as a graph in which each variable is a node and each correlation an edge. Both [10] ([10]) and [15] ([15]) used reduced partial correlation matrices of WAIS-IV test scores. The rational for the use of partial correlations is that one can rule out the relationship between any pair of variables as being due to other variables in the analysis and thus more readily infer causation ([6]). These partial correlation matrices were further reduced in the sense that “spurious or false positives” edges were removed by use of model selection procedures in the qgraph package from *R* (i.e., they created sparse correlation matrices with many values set to zero).

Both [10] ([10]) and [15] ([15]) compared model fit indices for network models with those for traditional latent variable models and found the former to provide a better fit. [10] ([10]) suggested that their results provided support for a mutualism model ([17]). According to [10] ([10]), mutualism assumes that reciprocal causation occurs among cognitive abilities during development. The positive manifold (i.e., the positive correlation between cognitive test scores) is due to these interactions rather than *g*. [15] ([15]) suggested that network models are compatible with process overlap theory ([11]). Process overlap theory holds that the positive manifold is due to multiple (mostly generalist executive) processes that determine performance on any specific test.

One problem in attempting to compare network models with traditional latent variable models of cognitive test batteries is that the two deal with different aspects of test correlations. The network models discussed used partial correlation matrices while latent variable models used uncorrected correlation matrices. It is possible to use uncorrected correlation matrix for network analysis, as shown in Table 2 of [10] ([10]), but this model is saturated (i.e., there are no degrees of freedom remaining after estimating model weights). As such, this model cannot be compared to models with residual degrees of freedom. Indeed, the network model is basically an alternative representation of the correlation matrix and the saturated model accounts for (i.e., is identical to) all of the covariance in the matrix. This is another reason that lesser edges need to be eliminated if comparisons with structural equation models are to be made.

From the description of the methods by [10] ([10]) and [15] ([15]), it is not clear what type of correlations were used to compare network and latent variable models. However, from the R code provided by [15] ([15]), it is clear that the fit of network models to partial correlations were compared to the fit of latent variable models to uncorrected correlation matrices. It is also apparent from this code and tabled data in these articles that the authors used the sparse correlation matrices only to identify the correlations to be included in the model and subsequently, estimated their magnitude to derive fit indices for comparison with latent variable models.

The present study was concerned with comparing the performance of a network model of the WAIS-IV standardization samples with several latent variable models. Both types of models were evaluated for their ability to fit both partial correlations and uncorrected correlations. In order to have a meaningful comparison of models, it is necessary that they be evaluated using identical data (i.e., correlation matrices). Otherwise, it is not possible to rule out the possibility that differences in model fit are due to the fact that disparate correlation matrices are compared.

## 2. Materials and Methods

### 2.1. Participants

All analyses were based on data from the standardization sample of the WAIS-IV ([18]). Three data sets were constructed consisting of data for individuals between 20 and 54 years of age (Tables A.3 through A.7, n = 1000), individuals between 16 and 19 years of age (Tables A.1 and A.2, *n* = 400), and individuals between 55 and 69 years of age (Tables A.8 and A.9, *n* = 400).

### 2.2. Procedures

Parameters were estimated from the 20–54-year-old data set (training set) and the other data sets (validation sets) were used for model validation with the parameter values fixed to the estimates obtained from the 20–54-year-old sample. Correlation matrices included all 15 subtests of the WAIS-IV. For each sample, tabled values were combined by first applying Fisher’s z transform, then averaging all of these values for each pair of subtests in a sample, and then taking the inverse transform to produce average r values. These procedures are identical to those used in [12] ([12]).

Partial correlations were computed using the cor2pcor procedure in the R package, corpcor. The result of this procedure is a matrix in which the correlation between each pair of variables has been adjusted for all other variables in the matrix. The network model was constructed with the EBICglasso in the R package qgraph. These procedures were used in the R code at the link to the OSF web site (https://osf.io/j3cvz/) referenced by [15] ([15]). All models were evaluated with both partial correlation and uncorrected correlation matrices.

All structural analyses were done with the SAS CALIS procedure ([14]) using default settings.

Latent factors were set equal to 1 as suggested by Anderson and Gerbing ([1]). Structural models included the four correlated factor model from Figure 5.2 of the WAIS manual (four group factor model) ([18]), a hierarchical model and a bi-factor model using the group-level factors from the Wechsler model, and a “penta-factor” model based on the WAIS-IV group level factors and four uncorrelated general factors ([12]). In addition, a network model was generated with the EBICglasso routine from the *q* graph package in *R*. The input to EBICglasso routine was a partial correlation matrix. 

The EBICglasso routine evaluates a series of L1 penalties (the graphical lasso) on the partial correlation matrix and selects the one that results in the lowest Bayesian Information Criterion (BIC) value. The graphical lasso simultaneously maximizes the log-likelihood of the data and minimizes (penalizes) the values of the estimated parameters (i.e., estimated correlation coefficients). The effect of the graphical lasso is to produce a sparse matrix in which a number of the off-diagonal elements are zero. The results of this procedure were used to select the correlations that were subsequently used in structural equation modeling for comparison to latent variable models. It should be noted that the EBICglasso model essentially considers the amount of covariance in the entire (partial) correlation matrix that can be accounted for by a subset of these correlations. 

## 3. Results

A summary of fit indices for models evaluating partial correlation matrices is shown in Table 1. The table shows *χ*^2^ values for the training set and validation sets. As can be seen in Table 1, the EBIClasso network produced the smallest values of *χ*^2^ in all cases, essentially replicating the effects seen by [10] ([10]) and [15] ([15]). Values for several fit indices (RMSEA, CFI and Akaike) presented in Table 1 also support this conclusion.

A summary of model fit indices for models fit to uncorrected correlation matrices is shown in Table 2. As can be seen in Table 2, the EBIClasso model provided the poorest fit of any of the models evaluated. The best fitting model in this series is the Penta-factor model.

Comparison of the baseline *χ*^2^ values for the partial and uncorrected correlation matrices provides insight into the markedly divergent performance of the EBIClasso model. In each case, the baseline model for partial correlations is less than 22% of that for the uncorrected correlation matrix. The partial correlations are the result of the removal of a large amount of covariance that is common to many of the WAIS-IV subtests. The sparse matrix produced by the EBIClasso model does not account for much of this covariance. 

## 4. Discussion

This study shows that a network model provides very good fit to matrices of partial correlations between the WAIS-IV subtests but very poor fit to matrixes of uncorrected correlations. The main issue then, is whether modeling partial correlations or uncorrected correlations is most appropriate.

[6] ([6]) discuss several reasons for favoring partial correlations in the analysis of networks. These include elimination of multicollinearity, identification of causal pathways, and identification of clusters in the network that correspond to latent variables. They further describe regularization of partial correlation matrices as a method to eliminate spurious edges. It should be noted that displaying regularized partial correlations as a network is simply a manner of visualizing the same data that could be examined in a table. As a result, what [6] ([6]) say about correlation networks applies to correlation matrices in general.

Partial correlation networks by definition rule out multicollinearity. Whether or not this is a desirable property of a correlation matrix depends on one’s perspective. [6] ([6]) wish to infer causality links in a network so that multicollinearity represents an interpretive problem. However, typical covariance structural modeling actually relies on multicollinearity to identify latent variables. The related point of inferring causation is more complex. The use of partial correlations eliminates the possibility that scores on a given variable are caused by other variables included in the partial correlation matrix. It is still possible that factors not included in the analysis are causal. Indeed, a common cause that is not directly measured is assumed to be the case in typical latent variable modeling. Other authors have pointed out that there are additional features of causal relationships, such as effects that are lagged in time, that require analyses of time series ([9]).

Conceptual problems arise from assuming that there are causal relationships between the observed scores on test batteries such as the WAIS-IV. This approach essentially requires that the test scores correspond in a one-to-one manner with the psychological traits that have causal relations. However, it seems unlikely that performance on a WAIS scale such as digit span directly causes performance on a WAIS scale such as arithmetic. At best, one would need to assume, following classical test theory, that the “true score” on a narrow trait exclusively indexed by digit span was causally related to the true score on a narrow trait exclusively related to arithmetic. However, such an approach would necessitate a proliferation of psychological traits associated with each of the various scales appearing on the multitude of cognitive test batteries that are currently in existence. The alternative is to assume that individual scales reflect the action of common latent traits. This argument is similar to that made by [13] ([13]) concerning the network analysis of symptoms in psychopathology. In this sense, the scores on WAIS sub-tests are analogous to the symptoms of psychopathology (i.e., they are symptoms of mental abilities). As discussed by [7] ([7]), it is certainly possible to have a network model of latent variables, but this is quite different from a network model with the nodes being scores on individual tests (i.e., manifest variables).

From a practical point of view, the fact that partial correlation matrices eliminate a major portion of the covariance between WAIS-IV test scales renders them inadequate for clinical interpretation. A model of WAIS partial correlations accounts for only a small fraction of the covariance in test scores, the majority of which is not unique to only two tests. Whether this common variance is best modeled with a single latent variable (i.e., *g*) and multiple group-level factors is a separate issue. Clinical interpretation should be based on models that account for a large portion of the actual variance in test scores. The same can be said for theoretical concerns.

Another issue concerns the role of regularization in model selection. The EBIClasso procedure provides a means of generating a sparse correlation matrix (i.e., a matrix where a portion of the elements are zero). Without some form of regularization, the model would essentially just be an alternative to the original partial correlation matrix, accounting for all of the covariance and having no residual covariance. [6] ([6]) state that “Even when two variables are conditionally independent, we still obtain nonzero (although typically small) partial correlations that will be represented as very weak edges in the network. These edges are called *spurious* or *false positives*”. This view seems to be based on the logic of null hypothesis significance testing. However, the authors of the graphical lasso used in the EBIClasso procedure state that it serves to increase sparisity and do not mention statistical significance or spurious values ([8]). In discussing the common features of diverse regularization procedures, [3] ([3]) suggest that they seek to solve the inherent stability problem when estimating a large number of model parameters. Regularization facilitates both the generalization of models to novel data as well as model interpretation due to greater simplicity. These considerations align with the methodology of predictive modeling rather than hypothesis testing ([16]). Sparse solutions have traditionally been obtained by heuristic procedures such as stepwise regression or elimination of eigenvectors with eigenvalues less than one in principal components analysis. More recently, regularization by methods such as parameter shrinkage by the Lasso used in the EBIClasso procedure have been advocated as alternatives. 

As noted earlier, [10] ([10]) suggested that their results provided support for a mutualism model ([17]), while [15] ([15]) suggested that network models are compatible with process overlap theory ([11]). From a latent variable perspective, the correlate factors model from the WAIS-IV manual and Table 1 and Table 2 of the present study might represent an appropriate form for the mutualism model. Likewise, the penta-factor model ([12]) could serve as a useful form for a latent variable model of process overlap theory. For example, the general form of the penta-factor model in the present study is consistent with [11] ([11]) conception of multiple overlapping domain-general executive cognitive processes being required for the performance of any given test item.

## 5. Conclusions

To summarize, network models based on manifest variables outperform latent variable models in accounting for the covariance in partial correlation matrices but do much worse when matrices of uncorrected correlations are considered. Since partial correlation matrices represent only a small fraction of the covariance in WAIS-IV test scores, they should not be the target of modeling.

## Figures and Tables

**Table 1 jintelligence-08-00007-t001:** Summary of fit indices for models evaluating WAIS-IV partial correlation. RMSEA is the root mean squared error of approximation, CFI is the comparative fit index, Gen-17 is the *χ*^2^ value for generalization of training weights to correlations from individuals between 16 and 19 years of age and Gen-63 is generalization of training weights to correlations from individuals between 55 and 69 years of age.

Model	*χ* ^2^	df	RMSEA	CFI	Akaike	Gen-17	Gen-63
Baseline	1707.50	105	-	-	-	638.15	720.08
Correlated Factors	465.17	82	0.0684	0.7609	301.17	383.36	331.03
Hierarchical	478.26	82	0.0696	0.7527	314.26	385.22	331.40
Bifactor	373.71	77	0.0621	0.7475	219.71	342.53	323.17
Penta-factor	109.69	33	0.0482	0.9521	43.69	320.38	288.17
EBIC Network	59.86	32	0.0295	0.9826	−4.14	310.91	315.56

**Table 2 jintelligence-08-00007-t002:** Summary of fit indices for models evaluating WAIS-IV Uncorrected Correlations. Abbreviations are the same as in Table 1.

Model	*χ* ^2^	df	RMSEA	CFI	Akaike	Gen-17	Gen-63
Baseline	8703.07	105	-	-	-	3015.39	3725.20
Correlated Factors	376.28	82	0.0599	0.9658	452.28	219.23	251.88
Hierarchical	483.96	82	0.0700	0.9444	559.96	247.02	282.74
Bifactor	277.36	75	0.0520	0.9765	367.36	186.48	226.43
Penta-factor	65.01	33	0.0312	0.9963	239.01	26.15	49.20
EBIC Network	1080.29	32	0.1811	0.8781	1016.29	507.39	754.63

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
