# Peer review of "The Effects of Using Partial or Uncorrected Correlation Matrices When Comparing Network and Latent Variable Models"

_jintelligence, 2020, doi:10.3390/jintelligence8010007_

Round 1

Reviewer 1 Report

The authors compare latent variable modeling with psychometric network analyses using the data of the WAIS-IV manual (Tables A1 through A7). While latent variable modeling uses the matrix of full correlations, the network modeling uses the observer variables and the partial correlation matrix. The network modeling is also related to the process overlap theory (POT) of intelligence. Both concepts (POT and network modeling) are rather new concepts and should therefore be explained in more detail for the reader. In addition, the results in Table 1 and 2 should also be represented as graphs, especially the EBIC network, in order to stress the differences in out come. results of network modeling are supposed to be represented graphically!

Author Response

I have added additional descriptions of mutualism and process overlap theory.  However, I have not included graphical representations of the models in Tables 1 and 2. There are five models in each table.  Also, the penta-factor model is rather difficult to represent graphically as there are many connections to all of the variables.  From a methodological standpoint, I believe that the comparisons in terms of model fit are most important. I have included references to original sources for these models.  

Reviewer 2 Report

This Brief Report concerns the use of partial correlations versus uncorrected correlation matrices when comparing network and latent variable models. Using the WAIS-IV standardization correlation matrices, the author compares the EBIC Network model with several latent variable models. The results show that the EBIC Network model provided best fit to the data only when partial correlation matrices were used; whereas the Pentafactor latent variable model provided best fit when uncorrected correlation matrices were analyzed. Based on the results, the use of network and latent variable approaches when modelling cognitive processes is then discussed. Essentially, the author points out critical issues associated with the use of network models.

This work provides good insight into methodological and theoretical issues that should be considered when judging network models. There are, however, some text passages that require editing in order to improve readability and clarity:

Page 2, lines 57-58: this sentence should be changed as the meaning is not clear.

Page 4, lines 134-137: this sentence is too long and the meaning is not clear.

Page 5, lines 188-189: Again, the sentence should be changes as the meaning is not clear.

Author Response

I have modified the first two sentences as suggested by the reviewer and eliminated the third. 

Round 2

Reviewer 1 Report

The added sentence in lines 60-63 should be rewitten. It does not make any sense: ...what these models are 57 useful in....

Author Response

I apologize for this error.  I had written an alternative ending for the introduction but it apparently did not make it into the last revision.  The new revision contains this.

Reviewer 2 Report

p. 2, lines 60-63: I do not see any modification in that sentence.

Author Response

I apologize for this error.  The correction did not make it into the last revision but I have included it in the present one.